# Using Light for Therapy of Glioblastoma Multiforme (GBM)

**DOI:** 10.3390/brainsci10020075

**Published:** 2020-01-31

**Authors:** Alex Vasilev, Roba Sofi, Ruman Rahman, Stuart J. Smith, Anja G. Teschemacher, Sergey Kasparov

**Affiliations:** 1School of Physiology Pharmacology and Neuroscience, University of Bristol, Bristol BS8 1TD, UK; otherlife@bk.ru (A.V.); rs16747@bristol.ac.uk (R.S.); anja.teschemacher@bristol.ac.uk (A.G.T.); 2Institute of Living Systems, Immanuel Kant Baltic Federal University, 236041 Kaliningrad, Russia; 3Faculty of Medicine, King Abdul-Aziz University, Alehtifalat st., Jeddah 21589, Saudi Arabia; 4Children’s Brain Tumour Research Centre, Nottingham Biodiscovery Institute, School of Medicine, University of Nottingham, Nottingham NG7 2RD, UK; ruman.rahman@nottingham.ac.uk (R.R.); stuart.smith@nottingham.ac.uk (S.J.S.)

**Keywords:** glioblastoma multiforme, photodynamic therapy, photosensitiser

## Abstract

Glioblastoma multiforme (GBM) is the most malignant form of primary brain tumour with extremely poor prognosis. The current standard of care for newly diagnosed GBM includes maximal surgical resection followed by radiotherapy and adjuvant chemotherapy. The introduction of this protocol has improved overall survival, however recurrence is essentially inevitable. The key reason for that is that the surgical treatment fails to eradicate GBM cells completely, and adjacent parenchyma remains infiltrated by scattered GBM cells which become the source of recurrence. This stimulates interest to any supplementary methods which could help to destroy residual GBM cells and fight the infiltration. Photodynamic therapy (PDT) relies on photo-toxic effects induced by specific molecules (photosensitisers) upon absorption of photons from a light source. Such toxic effects are not specific to a particular molecular fingerprint of GBM, but rather depend on selective accumulation of the photosensitiser inside tumour cells or, perhaps their greater sensitivity to the effects, triggered by light. This gives hope that it might be possible to preferentially damage infiltrating GBM cells within the areas which cannot be surgically removed and further improve the chances of survival if an efficient photosensitiser and hardware for light delivery into the brain tissue are developed. So far, clinical trials with PDT were performed with one specific type of photosensitiser, protoporphyrin IX, which tends to accumulate in the cytoplasm of the GBM cells. In this review we discuss the idea that other types of molecules which build up in mitochondria could be explored as photosensitisers and used for PDT of these aggressive brain tumours.

## 1. Introduction

The term glioma encompasses all tumours arising from the glia-like cells in the brain and spinal cord. High grade gliomas are the most common primary malignant brain tumours in adults and account for almost 80% of such tumours. The cell-of-origin of gliomas is still an open question and it is thought that they may develop from several types of progenitors, including neuronal stem cells (NSC), oligodendrocyte progenitor cells (OPCs) and astrocytes [1]. As with other cancers, factors leading to the tumour development are multiple and not easily defined, including age, genetic family predispositions, possibly exposure to ionising radiation, magnetic fields, pesticides, and solvents [2,3].

In 2016, the World Health Organization (WHO) classification of primary central nervous system (CNS) tumours was thoroughly revised and corrected, now taking into account the molecular characteristics of gliomas [4]. Regarding gliomas, they are graded based on histological features (anaplasia, atypia, proliferation index, and neovascularisation) into low grade gliomas (LGG) and high-grade gliomas (HGG). LGGs include grade 1 slowly proliferative gliomas and grade 2 infiltrative LGG. HGGs include grade 3 and grade 4 anaplastic infiltrative gliomas which are also referred to as glioblastoma multiforme (GBM) [5].

Molecular classification does not change grading, but rather helps in making decisions regarding the best treatment approach and predicting prognosis. According to the mutation status of the isocitrate dehydrogenase (IDH) gene, GBM can be either IDH wild-type or IDH-mutant. IDH mutation in GBM is frequently associated with TP53 mutation and has a generally better prognosis than IDH-wildtype glioblastoma [6]. O6-methylguanine-DNA methyltransferase (MGMT) is a DNA repair enzyme encoded by the gene MGMT located on chromosome 10q26. Methylation of the promoter of this gene is found in 35%–45% of HGG and is associated with a better response to chemotherapy by alkylating agent temozolomide (TMZ) and better overall survival [6]. About a third of paediatric GBM patients have mutations in TP53 and ATRX (Alpha thalassemia/mental retardation syndrome X-linked) genes [7]. In addition, mutations in the promoter of TERT, a gene that encodes the catalytic subunit of telomerase, are observed in a significant subset of GBM [8]. Other common molecular genetic alterations associated with GBM include: phosphatase and tensin homolog (PTEN) mutations, epidermal growth factor receptor (EGFR) amplification, cyclin-dependent kinase 4 (CDK4) amplifications, and cyclin dependent kinase inhibitor 2A (CDKN2-A) homozygous deletion [9]. Overall, GBM are highly heterogeneous in terms of their molecular makeup and this, combined with high genomic instability and intra-tumour variability, greatly reduces the chances of finding a “magic bullet” drug against this type of cancer. Further analysis of molecular diversity of GBM is outside of the scope of this review, but clearly, it will have implications for any potential therapy, be it pharmacological or alternative.

## 2. The Problem

Surgical removal, field radiotherapy (60 Gy) and subsequent systemic administration of temozolomide (TMZ) is the current standard-of-care for GBM. Whilst the bulk of the tumour can often be resected, GBM almost always re-occur due to the rapid proliferation of infiltrative residual tumour cells. New growth typically originates from the walls of the surgical cavity or parenchyma within <2 cm of the tumour margin [10]. This is due to small patches of GBM, the “scout” cells, which infiltrate the area adjacent to the tumour. Essentially in all cases, residual growth within critical regions such as critical white matter tracts or grey matter nuclei, cannot be surgically removed without a risk of severe neurological damage. In order to prevent or impair tumour recurrence and improve patient survival rates, it would be necessary to cleanse the cavity and the surrounding tissue. This could improve survival for patients but calls for a different approach, such as photodynamic therapy (PDT), using specific sensitising agents termed photosensitisers (PS).

## 3. Development of the Concept of PDT

Conceptually, PDT is the use of a chemical, which is able to absorb photons in order to trigger a process, detrimental for the tumour cells. Why should development of PDT for GBM be considered? The first obvious answer is that PDT, in principle, may be effective against some cancers, based on results of experiments in-vitro, rodent models, and clinical studies. Examples of cancers, where encouraging results with PDT have been obtained in-vitro include breast cancer [11], oesophageal cancer [12], liver carcinoma [13], and colorectal cancer [14]. Further information on in-vivo application of PDT can be found elsewhere [15]. Second, in spite of all the advancements in understanding of the molecular nature of GBM, the standard of care today (surgery, chemotherapy, and radiotherapy, known as Stupp protocol [16]) has not changed in the past 15–20 years and prognosis has not improved significantly since then. This indicates a crucial need for alternative, novel strategies of treatment, especially after molecular targeted therapies have thus far failed to show sufficient efficacy in clinical studies [17]. The third reason is that PDT does not (at least not directly) rely on a specific GBM marker or molecular signature. GBM cells are known for their genetic instability and intra-tumour heterogeneity which is the major reason for failure of targeted therapy in GBM. In contrast, preferential action of PDT on cancers usually relies on common, ubiquitous features of tumour cells or the tumour microenvironment such as immaturity of neovasculature, poor lymphatic drainage, and acidic pH [18]. Therefore, this therapeutic approach warrants consideration as it may circumvent intra- and inter-tumour genetic heterogeneity.

Apparently, the first attempt to use PDT was conducted by Friedrich Meyer–Betz in 1913 with photosensitiser hematoporphyrin [19]. Many later studies reported that injection of hematoporphyrin results in fluorescence of tumour cells. In 1960, a group of scientists from Mayo clinic developed the first PS, hematoporphyrin derivative (HpD), with a greater degree of tumour localisation [20]. At that point it was mainly used to diagnose tumours of the bladder, bronchus, and oesophagus. Later came the realisation that visualisation of the tumours is not the only possible application for the PS. In 1966, treatment of cancers using PS, now known as PDT, was first applied to breast cancer, using HpD. Since then, many studies have reported attempts of PDT with HpD (or its more active commercial form; photofrin II) in many types of cancers [20,21,22,23]. For brain tumours, PDT was first applied in 1972 when Diamond et al. [24] demonstrated phototoxicity of hematoporphyrin against gliomas in vivo and in vitro. Porphyrin PDTerapy delayed the growth of gliomas that were implanted in rats. Tumour growth was suppressed for 10–20 days, but eventually, viable areas from deeper regions of the tumours began growing again [24]. A selection of further in-vivo studies that attempted to develop PDT are listed in Table 1.

## 4. Principles of PDT

PDT requires a light source, a PS, and oxygen. The majority of PS reported in the literature are tetrapyrroles consisting of four pyrrole or pyrrole-like rings [39]. They can be classified according to their chemical structures into four major groups: porphyrin derivatives, chlorin derivatives, bacteriochlorin derivatives, and phthalocyanine derivatives [39]. The PS localises to target cells and, when subjected to light at a certain wavelength, absorbs the photon. This brings electrons into the excited state and eventually triggers photochemical reactions of two types. In the first type of reactions, interaction occurs directly with the molecules of the biological substrate, which ultimately leads to the formation of free radicals. In the second type, an excited PS interacts with an oxygen molecule to form singlet oxygen, which is cytotoxic to living cells since it is a powerful oxidising agent [40] (Figure 1). However, singlet oxygen only exists for <4 μs and migrates for a maximum of 1 micrometer, therefore not causing much harm to the adjacent healthy cells, or even organelles within the same cell [41]. Thus, the downstream consequences of PDT are directly associated with the pattern of intracellular accumulation of the PS.

Three main mechanisms are responsible for the PDT detrimental effect on tumours. Firstly, generation of reactive oxygen species (ROS) either in the cytoplasm or mitochondria directly damages proteins, lipds, DNA, and other molecules, resulting in cancer cell death mainly due to the activation of apoptosis. Second, some agents used for PDT can induce coagulation and thrombosis of microvessels, resulting in ischemia, necrosis and tumour infractions, this was noted in all studies with 5-aminolevulinic acid (5-ALA). Finally, PDT may activate immune cells and/or stimulate cytokines to elicit an immune response against tumour cells [19,42].

Photons absorbed by the PS transiently move electrons into higher energy orbits, changing their state from ground (S_O_) to excited (S_E_). Relaxation into the S_O_ can lead to fluorescence but also to the transfer of energy to molecular oxygen, which normally exists in so-called triplet state (^3^O_2_) where the two nuclei are sharing a pair of electrons. This leads to formation of highly reactive singlet oxygen (^1^O_2_) with a very short life span. The other family of products are reactive oxygen species (ROS) which are also highly reactive. In case of mitochondria, the key consequences are loss of mitochondrial membrane potential (MMP), opening of the mitochondrial transition pore (MPTP), leading to release of cytochrome C, activation of caspases and damage to the mitochondrial DNA. As the mitochondrial genome has very limited capacity for reparation, this eventually leads to the demise of the mitochondria and depletes the cell of energy. The mechanisms of PS activation depicted here are illustrated also in Akimoto [41], however mitochondria is somewhat special in that if a PS builds up there, mitochondrial DNA and proteins appear extremely close to the site of ROS and singlet oxygen production. We believe that, by targeting PDT to mitochondria, one can achieve a powerful PDT effect.

## 5. PDT of GBM with Porphyrins

To date, the vast majority of clinical approvals for PDT are for “superficial” or “luminal” pathologies, disorders of skin (e.g., actinic keratosis) and retina (e.g., age-related macular degeneration), and early bronchial or oesophageal cancer [43,44,45]. This is due to the simple fact that the physical principle of PDT is more applicable to cells located on the surface rather that deeper ones. In addition, local application of PS to the surface helps to avoid systemic toxicity. Several clinical trials are currently ongoing to evaluate PDT efficacy and safety in different types of cancers, including breast, prostate, lung, bile duct, and also GBM [46]. Photofrin, talaporfin, and 5-ALA have been used both in research and clinical trials [47]. Fluorescence induced by 5-ALA was used to detect infiltrative tumour during surgery by Stummer W (1998) [48]. Its use in florescence guided surgery is associated with greater rates of complete excision of tumour tissue and with significantly prolonged progression free survival and currently 5-ALA, under the trade name Gliolan^©^, is approved by the FDA and widely used for imaging tumour tissue including GBM [47]. Levels of florescence detected from an area via intra-operative imaging were strongly correlated with the histological presence of tumour tissue [47]. 5-ALA is an endogenous non-proteinogenic amino acid, which serves as a precursor of the fluorescent and mildly phototoxic protoporphyrin IX (PpIX) in the heme biosynthesis pathway. The conversion takes place in the mitochondria but PpIX is not retained there. PpIX appears to selectively accumulate in tumour tissue and emits red fluorescence when illuminated by deep blue light, while normal brain tissue is not fluorescent [48,49,50].

Work by two groups needs particular attention in the context of using 5-ALA and PpIX for PDT of GBM. German scientists [32,37,47,51] have explored this modality in experimental and clinical settings and combined it with the stereotaxic placement of light delivery fibres into the brain. Moreover, they even used the same fibre optics to measure fluorescence from the brain tissue. In their work they chose to activate PpIX by red light (635 nm) delivered from a laser source. The choice of such long wavelength was motivated by the better penetration of red light in the brain tissue but clearly the efficiency of excitation was reduced since the absorption peak of PpIX is 402 nm, which is at the lower limit of visible light, while its peak of emission is 631 nm [52,53] although Mahmoudi et al. [47] have also identified a minor peak at 635 nm. In these studies, substantial destruction of tumour tissue was observed in conjunction with signs of vascular damage and necrosis. This could reflect a rather large amount of energy dissipated in the brain tissue. For example, in the Beck et al. 2007 study [32], some patients received over 11,500 J of light energy. In fact, laser application alone caused the same degree of tissue destruction in the rat model, with and without loading with 5-ALA [51]. The other group from Japan [53,54], also actively looked into the use of porphyrins in GBM. Overall, their approaches are similar to what was explored in studies mentioned above [31,37,47,51]. Possible therapeutic benefits were noticed in some patients but overall, no convincing conclusions have been reached and the attention more recently shifted to further perfecting the use of 5-ALA and porphyrins for intra-operative GBM diagnostics [53]. It is also not entirely clear whether the tumour destruction noted in some of these studies was due to the genuine PD effect (e.g., generation of ROS or singlet oxygen) or rather due to a less specific effects of heat which caused thrombosis and destruction of blood vessels. In 2017 a clinical trial was initiated to evaluate the use of 5-ALA as a PDT agent (INDYGO, NT number: NCT03048240) where illumination of 5-ALA-stained tissue is added to the conventional standard of care protocol. The results have not yet been posted and evidence for added value of this approach is currently still lacking [55]. Another trial with 5-ALA, which is currently registered at ClinicalTrials.gov (NCT03897491) by Photonamic GmbH&Co.KG, “PD L 506 for Stereotactic Interstitial Photodynamic Therapy of Newly Diagnosed Supratentorial IDH Wild-type Glioblastoma”, is not yet recruiting.

GBM is not the only possible application for 5-ALA. For example, it has tested for treatment of tumours of the gastrointestinal tract by J Regula et al. [56]. PDT resulted in superficial necrosis of the mucosa in the areas exposed to light [56]. In vitro study in malignant paediatric tumour cell lines including medulloblastoma (DAOY, UW228), pNET (PFSK-1) and rhabdoid tumour (BT16) lines, demonstrated that PDT with 5-ALA (50 μg/mL) resulted in death of tumour cells, but not of control cells (in absence of 5-ALA or laser irradiation) [57].

## 6. Technical Aspects of Light Delivery into the Brain

Hardware for PDT in the brain has been reviewed elsewhere [58]. Briefly, light sources could be of non-coherent (various lamps and light emitting diodes) and coherent (laser) type. From the practical point of view, beams from a lamp cannot be sent down a fine optical fibre (less than a few mm^2^) while lasers allow exactly that and light emitting diodes being somewhere in the middle. For this reason, light from the bulbs, such as used for intraoperative visualisation of 5-ALA staining is typically delivered via liquid light guides with the cross section of >1 cm^2^. Obviously, such thick objects cannot be inserted into brain tissue without causing damage.

Control of light intensity and availability of pulsatile mode are also limited when using lamps and require additional optical elements, while solid state lasers often can be controlled directly using TTL-pulses or regular PC-compatible interfaces, such as USB ports. The spectrum of the light source is also important, specifically, longer wavelengths (infra-red part of the spectrum) are a direct source of heat and this may be very damaging to healthy brain parenchyma.

A very interesting approach is the interstitial PDT (iPDT) implemented in Beck et al. [32] and Johansson [37], studies where light is delivered inside the brain tissue rather than applied to the surface. Clearly, simply increasing the power dissipated directly from the tip of a fibre is not the way forward because firstly, a narrow beam does not allow to saturate the volume of tissue with light. Secondly, any wavelength which can realistically excite fluorescence (e.g., <650 nm) gets absorbed within the first few mm [47] where the energy is dissipated as heat. Hence the PDT may easily become a heat shock to the brain, which is bound to cause necrosis, oedema and perhaps vascular thrombosis with potentially risky consequences. It is therefore very interesting that Beck et al. [32] have used cylindrical light diffusers, which allowed to illuminate a substantial volume of brain tissue from an array of such optical elements [37]. These assemblies were stereotactically placed under X-ray control into the brain of patients which was possible due to the presence of special X-ray markers on them. Overall, this type of hardware seems to offer advantages. These studies also prove that placement of several fine optical fibres into the tissue of the brain is possible and is well tolerated.

The wavelength of light is of principle importance. For instance, tissue penetration of 630 nm is 3–4 times greater than of 400 nm and, in general, the longer the wavelength of light, the better it spreads. On the other hand, shorter wavelength photons carry much more energy and are more potent as exciters. For example, a blue photon at 402 nm carries 4.94 × 10^−19^ joules while a red photon at 623 nm - 3.14 × 10^−19^. In other words, a 402 nm photon carries ~1.6 times more energy. This has direct implications for the type of reactions such photons can initiate in the molecules which absorb them.

Interestingly, according to the mathematical model, singlet oxygen yield is the highest at 510 nm, it is similar at 532 nm and 488 nm, but very low at 578 nm and 630 nm. This factor could counterbalance the less efficient penetration of light of <600 nm [58], potentially indicating that the optimum wavelength lies somewhere within the range of 510–550 nm, which is yellow-green light. This consideration is supported by the known ability of many endogenous molecules (for example, flavonoids and porphyrins) to absorb blue light (<500 nm), which inevitably limits penetration and leads to greater off-target effects [59]. In other words, the optimum wavelength should, first, allow reasonable light penetration (>500 nm) and, second, potently generate singlet oxygen (490–550 nm). Finally, infra-red light (>700 nm) when absorbed, only generates heat and can easily cause thermal damage.

## 7. Challenges for Development of PDT for GBM

Laboratory research on application of light for therapy of GBM has not yet produced conclusive results but this does not mean that this strategy is not viable. Most, if not all, PDT experiments on gliomas used commercially available glioma cell lines [60,61,62,63,64,65,66,67]. As is the case with many cancer cell lines, there is always uncertainty about the origin of these cells, about possible contamination with other cell lines (e.g., HeLa cervical cancer cells, a common contaminating cell), and the effect of culturing which may lead to the accumulation of mutations with every cell division [68]. For example, in a recent study, commonly used glioma cell lines were tested for DNA fingerprints. Among the 39 lines tested, 3 lines were found to be misidentified as glioma lines and two more lines were found to be cross-contaminated [69]. In addition, short tandem repeat profiling of two U87 cell lines purchased from two different companies revealed non-identical origin of these cells and only one of them had a genotypic profile identical to the U87 profile published in Cell Lines Service [70]. All this casts a shadow of doubt on the pathophysiological relevance and accuracy of representation of many of the in vitro cell models. As a consequence, the relevance of the reported effects of PDT may be questioned. Therefore, it is much preferable to utilise only primary, patient-derived and validated GBM cell lines, especially those isolated from the invasive margin of GBM [71].

There is also an important constraint imposed by the involvement of oxygen in photodynamic reactions. Hypoxia is an acknowledged feature of the GBM microenvironment. In vitro studies usually neglect this fact and are hardly ever performed under conditions of limited oxygen levels to mimic the GBM microenvironment. This will result in over-estimation of PDT effectiveness and inability to accurately predict its efficacy in patients. Even though, in real life, PDT for GBM could be applied directly to the exposed tumour bed after resection, the previous state of chronic hypoxia and the resultant overexpression of hypoxia inducible factor 1 (HIF-1) and downstream effectors in GBM may alter the tumour cells’ response to oxygen-dependent killing mechanisms. To aid saturation of tissue with oxygen, Beck et al. 2007 [32] ventilated patients with 100% O_2_. However, some researchers argue that type 1 photodynamic reactions (unlike type 2) are not oxygen-dependent and would still allow PDT effect to develop in case of hypoxic conditions [72].

PDT in published clinical studies on GBM typically consisted of surgery with or without PDT (test arm vs. control arm). These early phase studies suggested a survival advantage for test arms compared to controls. However, side effects after PDT were severe and included deep venous thrombosis, neurological deficits, and multi-organ failure [33,35,53,73]. Although PDT was perhaps effective, it is very difficult to distinguish whether reported side effects are caused by the surgery or the PDT, or simply complications of the disease itself. The small sample size in these clinical studies further complicates the interpretation of results. Brain oedema could be controlled by decreasing the light dose or by applying the emerging concept of metronomic PDT, in which PS and light are given in continuous, low-dose fashion over extended periods of time. This was found to decrease collateral damage, coagulation, necrosis and/or oedema of the surrounding tissue [74], possibly due to the lower chances of an irreversible thermal damage.

From the available literature, it appears that so far, the efficacy of porphyrins as exciters for PDT remains uncertain, although they greatly aid the detection of tumour margins. In fact, it is hard to distinguish whether the damage to the tumour caused in the animal or human experiments was due to the porphyrins at all. For example, in previously mentioned Olzowy et al. experiments [51], red (633 nm) light caused the same degree of damage irrespective of the loading with 5-ALA. It seems that a different combination of the photosensitiser and wavelength could produce better results, possibly via not focusing only on selective biotransformation of 5ALA in the GBM but exploiting some other PS and features of the tumour cells.

The other area of important development is the hardware for light delivery. The important key steps have already been made by introduction of the interstitial PDT. This method seems to be the only feasible strategy to deliver light to the GBM cells underneath the surface of the cavity left after the debulking. Stereotaxic placement of the arrays of special light diffusers may help to illuminate the required volume of the parenchyma without causing dangerous mechanical damage to the surrounding tissue.

## 8. Mitochondria as a Target for PDT

Although the idea to target mitochondria for cancer therapy is not new, it has not been yet convincingly demonstrated in the context of PDT. Nevertheless, some studies suggest that this may be a feasible strategy.

The best known group of mitochondrially targeted PS are the “delocalised lipophilic cations” such as rhodamine-123 (Rhod-123), known for its ability to accumulate in mitochondria. Its absorbance peaks at 530 nm (green light). Castro D.J. et al. used human carcinoma cells loaded with Rhod-123 (1 μg/mL for 1 h) and demonstrated that PDT affected cell division but not cell viability [75]. The authors argued that the effect was achieved by the photodynamic effect and not temperature increase. Energy density was <950 J/cm^2^ and temperature was kept below 40 °C. Powers et al. used cultured glioma line U-251MG. Cells were incubated for 30 min with Rhod-123 (10 µg/mL) and then irradiated using variable power densities (10 to 300 mW/cm^2^) and variable energy densities (0 to 200 Joules/cm^2^) of blue-green light between 488 and 514 nm, using a continuous-wave argon laser [76]. A cytotoxic effect was observed when power densities of light were 100 mW/cm^2^. However, with an increase in exposure time, a logarithmic decrease in cell survival occurred, indicating that the photochemical effect depended on the concentration of Rhod-123 and the duration of exposure to light much more than the intensity of light [76]. There is some evidence that carcinoma cells have an increased rhodamine-123 uptake [77] but it is not clear why this is or whether this is the case for other tumours, such as GBM. It should be noted that although Rhod-123 can accumulate in mitochondria, it is certainly not specifically localised there, and sulforhodamine 101, commonly used to stain astrocytes in-vivo, gives strong cytoplasmatic fluorescence [78].

The other group of PS that could affect the mitochondria are porphyrin derivatives, including first generation PS HpD and photofrin. The first clinical application was demonstrated by Perria et al. in 1980 with HpD [25]. HpD act on mitochondria to reduce cytochrome C activity [79] and affect mitochondrial respiration, oxidative phosphorylation, and concentration of Ca^2+^ [80]. Second generation photosensitisers include Temoporfin (trade name Foscan). In myeloid leukaemia cells, temoporfin was reported to be present in mitochondria and light activation with constant light (band of approximately 500–800 nm) caused mitochondrial damage and the release of cytochrome C [81]. At the same time, Temoporfin clearly accumulated in a wide range of cellular structures and vesicles and therefore cannot be seen as a mitochondria-selective PS. Whether any of these PS can be selective to GBM cells compared to healthy tissue is not clear. The cellular effects caused by photoactivation of 5-ALA mentioned above are not well studied and might or might not be related to mitochondrial damage. Moreover, 5-ALA can induce Ca^2+^-dependent swelling of the mitochondria even without photoactivation [82].

## 9. Conclusions

The success of PDT will depend on the ability to selectively damage GBM cells compared to normal constituents of the brain such as astrocytes and neurones. How can this be achieved? We believe that the mitochondrial membrane potential (MMP) could be used to preferentially build up PS in GBM mitochondria. MMP is generated by proton pumps (complexes I, III, IV) which create an electrochemical gradient between cytoplasmic and inner sides of the membrane. This transmembrane electrical gradient together with proton gradient form the transmembrane potential of hydrogen ions which is essential for oxidative phosphorylation reactions and synthesis of ATP in the cell [83]. It is known that the mitochondrial potential of healthy cells ranges from −108 and −159 mV, with a mean value of −139 mV while mitochondria in cancer cells are much more hyperpolarised and have a potential value around −210 mV [84]. Importantly, hyperpolarisation of mitochondria seems to be a generic landmark feature of tumour cells, including GBM. Therefore, there is a possibility that cationic PS, which are driven into the mitochondria by their membrane potential could open a therapeutic window by preferentially damage GBM cells over brain glia or neurones.

It is also important to deliver sufficient light energy into the tissue while avoiding use of the blue/deep-blue-UV spectrum or using light sources which significantly heat up tissue such as the infra-red part of the spectrum. Critically, the light needs to be delivered into the parenchyma because even the most generous estimates of light penetration into brain tissue are in the range of single millimetres while, in order to cleanse the tumour infiltration from the wall of the surgical cavity, light needs to reach PS in cells at depths of up to 2 cm. As mentioned above, this idea has already been implemented by W. Stummer’s and F.W. Kreith’s groups [32] and also by K. Shimizu and colleagues [53]. Moreover, light can be delivered via multiple optical fibers with cylindrical diffusers, so that the collective array effectively excites a significant volume of tissue [32]. Mathematical models and further hardware development will help to further improve this critical aspect of PDT [85].

## 10. Perspectives

GBM remains one of the deadliest cancers known and there is an acute need for new methods of therapy. Primary tumours often localise relatively superficially in the brain and can be removed. Therefore, it is almost surprising that such surgical treatment combined with radio therapy and temozolomide, which is the current standard of care, is so unsuccessful. The key reason for the low survival rates of patients with GBM is local recurrence, which happens in the vast majority of cases and is caused by the infiltrating GBM cells. Although surgical techniques such as fluorescence-guided surgery has allowed intra-operative real-time visualisation of tumour cells, infiltrating GBM cells cannot be mechanically removed, in most cases, because they spread into areas which are too risky to surgically destroy, for example critical white matter tracts. Therefore, the task for the future is to develop effective strategies to cleanse the infiltrating GBM cells from the parenchyma immediately adjacent to the surgical wound. It is here PDT could find its niche.

## Figures and Tables

**Figure 1 brainsci-10-00075-f001:**
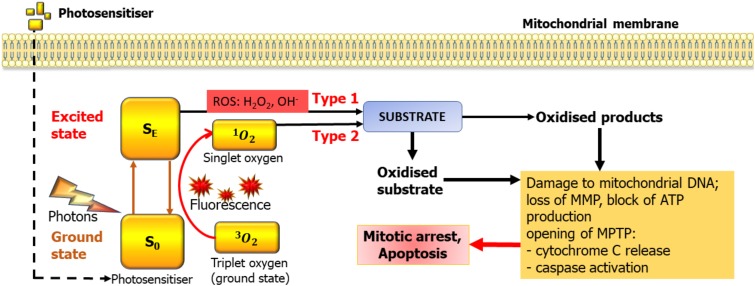
Basic mechanisms of photodynamic therapy (PDT) and its potential targets in mitochondria. MMP—Mitochondrial membrane potential, MPTP—mitochondrial transition pore.

**Table 1 brainsci-10-00075-t001:** A selection of in-vivo studies which utilised photodynamic therapy (PDT) for glioblastoma multiforme (GBM) therapy.

Reference	Number of Treated Glioblastomas	Photosensitiser				Light		Median Survival Time (Months)
		Trade Name	Agent	Dose	Localisation	Wavelength (nm)	Energy Density (J/cm^2^)	
Perria et al. (1980) [25]	Not defined	Photofrin^®^	Hematoporphyrin derivative (HpD)	5 mg/kg	Cell membrane; mitochondria	628	720–2400	6.9
Povovic et al. (1995) [26]	Newly diagnosed, 38 Recurrent, 40	Photofrin^®^	Hematoporphyrin derivative (HpD)	2–2.5 mg/kg	Cell membrane; mitochondria	628	70–260	Newly diagnosed, 24 Recurrent, 9
Rosenthal et al. (2003) [27]	Newly diagnosed, 7 Recurrent, 9	BOPP	Boronated porphyrin	0.25–8 mg/kg	Various, including mitochondria	630	25–100	Newly diagnosed, 7 Recurrent, 11
Stylli et al. (2005) [28]	Newly diagnosed, 31 Recurrent, 55	Photofrin^®^	Hematoporphyrin derivative (HpD)	5 mg/kg	Cell membrane; mitochondria	628	70–240	14.3
Muller and Wilson, (2006) [29]	Newly diagnosed, 12 Recurrent, 37	Photofrin^®^	Hematoporphyrin derivative (HpD)	2 mg/kg	Cell membrane; mitochondria	630	Mean, 58 Maximum, 150	7.6
Kostron et al. (2006) [30]	Recurrent, 26	Foscan^®^	mTHPC metatetrahydroxyphenylchlorin	0.15 mg/kg	Golgi apparatus, endoplasmic reticulum, and mitochondria	630	20	Recurrent, 8.5
Stepp H. et al. (2007) [31]	Newly diagnosed, 20	Levulan	5-ALA	20 mg/kg	mitochondria	633	200 mW/cm^2^	Newly diagnosed, 15.2
Beck T.J. et al., (2007) [32]	Recurrent GBM, 10	Levulan	5-ALA	20 mg/kg	mitochondria	633	Total 4320–11,520 J (200 mW/cm)	15
Eljamel (2008) [33]	Newly diagnosed, 13	Photofrin^®^	Hematoporphyrin derivative (HpD) 5-ALA	2 mg/kg	Cell membrane; mitochondria	630	100 × 5 sessions	Newly diagnosed, Mean survival 52.8 weeks
Kaneko et al., (2008) [34]	Total, 35	Photofrin^®^	Hematoporphyrin derivative (HpD)	2 mg/m^2^	Cell membrane; mitochondria	630	180	Total, 20.5
Akimoto et al. (2012) [35]	Newly diagnosed, 4 Recurrent, 6	Laserphyrin^®^	Talaporfin sodium	40 mg/m^2^	-	664	27	Newly diagnosed, 31 Reccurent, 9
Lyons M. et al. (2012) [36]	Total, 73	Photofrin^®^	Hematoporphyrin derivative (HpD)	2 mg/kg	Cell membrane; mitochondria	630	100	Mean survival 62.9 weeks
		Levulan	5-ALA					
Johansson et al. (2013) [37]	Total, 5	Levulan	5-ALA	20–30 mg/kg	Cell membrane; mitochondria	630	720	-
Muragaki et al. (2013) [38]	Newly diagnosed, 13	Laserphyrin^®^	Talaporfin sodium	40 mg/m^2^	-	664	27	Newly diagnosed, 24.8

5-ALA, 5-aminolevulinic acid.

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
