# Peer review of "Using Light for Therapy of Glioblastoma Multiforme (GBM)"

_brainsci, 2020, doi:10.3390/brainsci10020075_

Round 1
Reviewer 1 Report
The manuscript titled “Using light for therapy of glioblastoma multiforme (GBM)” is a mini review briefly covering the historical and possible future applications and challenges facing the use of PDT for GBM treatment. The review is well written and covers the major clinical trials as well as some of the limitations involving light distribution and the normal pre-clinical models used. Following this the authors presented ideas for future exploration that could improve the therapeutic response to PDT. With that being said, I am not sure that there is anything presented in the paper that is really novel or adds to the field substantially although I am aware of the limits placed on the authors. Secondly some of the conclusions, specifically surrounding light distribution and the available light treatments should be re-worked to better reflect the current state of the field.
With that being said below are more specific points:
Page 4, Table 1 – Overall this table is thorough, one missing clinical trial stands out: A. Johansson, F. Faber, G. Kniebuhler, H. Stepp, R. Sroka, R. Egensperger, W. Beyer and F. W. Kreth, Lasers in surgery and medicine, 2013, 45, 225-234.
Page 7, Line 148 – The authors present a good overview of some of the challenges with in vitro PDT experiments. However, no in vivo experimental data was critiqued or examined. It would further highlight some of the difficulties with examining the topic fully.
Page 8, Line 97 – The technical considerations section is quite limited and doesn’t take into account some of the new developments or covers what was previously used in many of the trials including diffuse light sources, stereotactic light placement, as well as tumour-light distribution modeling.
Page 8, Lines 200-202 – The paper cited, and the mathematical model used for wavelength choice is based on skin where both damage to the dermis was to be avoided and an organ which has different optical properties compared to brain. The authors should look cite and examine any relevant material for brain. Specifically, examining PDT dosimetry models.
Page 9, Line 271 – 274 – Light delivered as a cylinder or sphere has previously been demonstrated in clinical trials. Furthermore, other groups have used stereotactic fibre placement to achieve overall light distribution. This section should include some of this work including the paper cited to better reflect some of the method used during treatment.
Author Contributions Section – This section has not been completed not sure if this per author instructions or just missed.
Author Response
We would like to thank all the reviewers and the academic editor for their extremely helpful criticisms and suggestions. We have made multiple extensive changes and amendments and hope our revision is satisfactory.
Below we cite the criticisms and provide responses (bold) and extracts from the new text in red.
Page 4, Table 1 – Overall this table is thorough, one missing clinical trial stands out: A. Johansson, F. Faber, G. Kniebuhler, H. Stepp, R. Sroka, R. Egensperger, W. Beyer and F. W. Kreth, Lasers in surgery and medicine, 2013, 45, 225-234.
Apologies, this was an oversight on our part. This and other relevant papers from the Munchen group are now cited in several parts of the review and the Table 1.
Page 7, Line 148 – The authors present a good overview of some of the challenges with in vitroPDT experiments. However, no in vivo experimental data was critiqued or examined. It would further highlight some of the difficulties with examining the topic fully.
We have made multiple changes to the text, also added these paragraphs to the “Basic principles” and “Challenges” sections.
Legend:
Photons absorbed by the PS transiently move electrons into higher energy orbits, changing their state from ground (SO) to excited (SE). Relaxation into the SO can lead to fluorescence but also to the transfer of energy to molecular oxygen, which normally exists in so-called triplet state (3O2) where the two nuclei are sharing a pair of electrons. This leads to formation of highly reactive singlet oxygen (1O2) with a very short life span. The other family of products are reactive oxygen species (ROS) which are also highly reactive. In case of mitochondria, the key consequences are loss of mitochondrial membrane potential (MMP), opening of the mitochondrial transition pore (MPTP), leading to release of cytochrome C, activation of caspases and damage to the mitochondrial DNA. As the mitochondrial genome has very limited capacity for reparation, this eventually leads to the demise of the mitochondria and depletes the cell of energy. The mechanisms of PS activation depicted here are illustrated by Akimoto [27], however mitochondria is somewhat special in that if a PS builds up there, mitochondrial DNA and proteins appear extremely close to the site of ROS and singlet oxygen production. We believe that by targeting PDT to mitochondria one can achieve a powerful PDT effect.
Work by two groups needs particular attention in the context of using 5-ALA and PpIX for PDT of the GBM. German scientists [37] [38][39] [33] have explored this modality in experimental and clinical settings and combined it with the stereotaxic placement of light delivery fibres into the brain. Moreover, they even used the same fibre optics to measure fluorescence from the brain tissue. In their work they chose to activate PpIX by red light (635 nm) delivered via fibres from a laser source. The choice of such long wavelength was motivated by the better penetration of red light in the brain tissue but clearly the efficiency of excitation was reduced since the absorption peak of PpIX is 402 nm, which is at the lower limit of visible light, while its peak of emission is 631 nm [40] [41], although Mahmoudi et al [33] have also identified a minor peak at 635 nm. In these studies, substantial destruction of tumour tissue was observed in conjunction with signs of vascular damage and necrosis. This could reflect a rather large amount of energy dissipated in the brain tissue, for example in the Beck et al 2007 study [37] some patients received over 11,500 J of light energy. In fact laser application alone caused the same degree of tissue destruction in the rat model, with and without loading with 5-ALA [38]. The other group from Japan [42] [41], also actively looked into the use of porphyrins in GMB. Overall, their approaches are similar to what was explored in studies mentioned above [38][43][39][33]. Possible therapeutic benefits were noticed in some patients but overall, no convincing conclusions have been reached and the attention more recently shifted to the further perfecting the use of 5-ALA and porphyrins for intra-operative GBM diagnostics [41]. It is also not entirely clear whether the tumour destruction noted in some of these studies was due to the genuine PD effect (e.g. generation of ROS or singlet oxygen) or rather due to a less specific effects of heat which caused thrombosis and destruction of blood vessels. In 2017 a clinical trial was initiated to evaluate the use of 5-ALA as a PDT agent (INDYGO, NT number: NCT03048240) where illumination of 5-ALA-stained tissue is added to the conventional standard of care protocol. The results have not yet been posted and evidence for added value of this approach is currently still lacking [44].
GBM is not the only possible application for 5-ALA. For example, it has tested for treatment of tumours of the gastrointestinal tract by J Regula et al [45]. PDT resulted in superficial necrosis of the mucosa in the areas exposed to light [45]. In vitro study in malignant paediatric tumour cell lines including medulloblastoma (DAOY, UW228), pNET (PFSK-1) and rhabdoid tumour (BT16) lines, demonstrated that PDT with 5-ALA (50 μg/ml ) resulted in death of tumour cells, but not of control cells (in absence of 5-ALA or laser irradiation) [46].
Page 8, Line 97 – The technical considerations section is quite limited and doesn’t take into account some of the new developments or covers what was previously used in many of the trials including diffuse light sources, stereotactic light placement, as well as tumour-light distribution modeling.
We have revised this section and included the studies the reviewer has alluded to. It was an oversight on our part and thanks for pointing it out.
A very interesting approach is the interstitial PDT (iPDT) implemented in Beck et al [37] and Johansson [39] studies where light is delivered inside the brain tissue rather than applied to the surface. Clearly, simply increasing the power dissipated directly from the tip of a fibre is not the way forward because firstly, a narrow beam does not allow to saturate the volume of tissue with light. Secondly, any wavelength which can realistically excite fluorescence (e.g. <650 nm) gets absorbed within the first few mm [33] where the energy is dissipated as heat. Hence the PDT may easily become a heat shock to the brain, which is bound to cause necrosis, oedema and perhaps vascular thrombosis with potentially risky consequences. It is therefore very interesting that Beck et al [37] have used cylindrical light diffusers, which allowed to illuminate a substantial volume of brain tissue from an array of such optical elements [39]. These assemblies were stereotactically placed under X-ray control into the brain of patients which was possible due to the presence of special X-ray markers on them. Overall this type of hardware seems to offer advantages for light delivery into the brain. These studies also prove that placement of several fine optical fibres into the tissue of the brain is possible and is well tolerated.
Page 8, Lines 200-202 – The paper cited, and the mathematical model used for wavelength choice is based on skin where both damage to the dermis was to be avoided and an organ which has different optical properties compared to brain. The authors should look cite and examine any relevant material for brain. Specifically, examining PDT dosimetry models.
Corrected
Page 9, Line 271 – 274 – Light delivered as a cylinder or sphere has previously been demonstrated in clinical trials. Furthermore, other groups have used stereotactic fibre placement to achieve overall light distribution. This section should include some of this work including the paper cited to better reflect some of the method used during treatment.
Thanks for pointing out, we have included this paragraph in the “Technical aspects” section
A very interesting approach is the interstitial PDT (iPDT) implemented in Beck et al [37] and Johansson [39] studies where light is delivered inside the brain tissue rather than applied to the surface. Clearly, simply increasing the power dissipated directly from the tip of a fibre is not the way forward because firstly, a narrow beam does not allow to saturate the volume of tissue with light. Secondly, any wavelength which can realistically excite fluorescence (eg <650 nm) gets absorbed within the first few mm [33] where the energy is dissipated as heat. Hence the PDT may easily become a heat shock to the brain, which is bound to cause necrosis, oedema and perhaps vascular thrombosis with potentially risky consequences. It is therefore very interesting that Beck et al [37] have used cylindrical light diffusers, which allowed to illuminate a substantial volume of brain tissue from an array of such optical elements [39]. These assemblies were stereotactically placed under X-ray control into the brain of patients which was possible due to the presence of special X-ray markers on them. Overall this type of hardware seems to offer advantages for light delivery into the brain. These studies also prove that placement of several fine optical fibres into the tissue of the brain is possible and is well tolerated.
The wavelength of light is of principle importance. For instance, tissue penetration of 630 nm is 3-4 times greater than of 400 nm and, in general, the longer the wavelength of light, the better it spreads. On the other hand, shorter wavelength photons carry much more energy and are more potent as exciters. For example, a blue photon at 402 nm carries 4.94×10^-19 joules while a red photon at 623 3.14×10^-19, in other words 402 nm light carries ~1.6 times more energy. This has direct implications for the type or reactions such photons can initiate in the molecules which absorb them.
Author Contributions Section – This section has not been completed not sure if this per author instructions or just missed.
COMPLETED
Reviewer 2 Report
The authors made an up-to-date mini-review regarding the use of light in the field of GBM treatment. They exactly acknowledge the definite need for fundamental breakthrough in treating GBM on top of surgical removal. There have been various approaches for this goal, and light or photo-mediated therapy should be one of the promising and indispensable. This review nicely describes the clinical history, reasons of current treatment, technical aspect, and scientific background behind the scene. Thus, this review poses a good insight and gives accessible information for readers, particularly clinicians, as well as basic researchers. I would endorse publication of the manuscript if the authors can provide the following modifications.
Major
Figure 1 resembles too much the figure in Akimoto J. Photodynamic Therapy for Malignant Brain Tumors. Neurologia medico-chirurgica. 2016;56(4):151-7, which is not cited in the manuscript. I would not say this is a plagiarism, but might be suspected. I would recommend modification of the figure or making up from scratch, so that the figure can convey the true meaning that authors may want to show. Line 250; It is reasonable that the authors focuses on mitochondrial effect of PDT if they specialize in the field. Still, if this is a review for PDT of GBM, the use of talaporfin sodium could be considered to be described, because this is the only government-approved PDT for GBM so far. Line 274-283; This summary well describes the reason for need for PDT, but there is too little about PDT itself. Some key text should be added, at least to match the abstract.Minor
Title; as the authors focused very much on the function of mitochondria, this point could be included in the title, so that the potential readers who are particularly interested in those function in (and relationship to) PDT. Line 17; in explanation of photosensitizer, its specific accumulation to tumor should be described, although referring to molecular finger print is also important. (“not specific to particular molecular fingerprint” could be misleading, although correct, if the readers are expecting its particular accumulation to cancer cells.) Line 33; the description about grading system in WHO criteria are not correct and needs citation, which is later cited well. The criteria were determined to match clinical outcome. Also, grade 1 glioma should be included in low grade glioma (LGG). Table 1, Row 1 (for Perria et al); “Not defineds” probably a typo. Table 1, Last row; citation is missing for Akimoto el al. Line 139; surgery “by” Stummer should be correct. Line 140; “Enhancing tumor” might be difficult to understand for researchers. MRI enhancement effect should be mentioned briefly. Line 141; the comma between survival and currently should be a period instead. Also, FDA approval of 5-ALA was rather late compared to the status in Europe and Japan. This fact should be mentioned. Line 245; In terms of clinical trial using 5-ALA for PDT, INDYGO is not the only one. Interstitial PDT should be mentioned as a review. Please check “PD L 506 for Stereotactic Interstitial Photodynamic Therapy of Newly Diagnosed Supratentorial IDH Wild-type Glioblastoma”. Line 268-271; This part is critically important for clinical application of PDT. Since the authors have made Technical aspect subheading, this description could be moved there so to help readers understand the situation. The following future prospective are reasonable to be described here.
Author Response
We would like to thank all the reviewers and the academic editor for their extremely helpful criticisms and suggestions. We have made multiple extensive changes and amendments and hope our revision is satisfactory.
Below we cite the criticisms and provide responses (bold) and extracts from the new text in red.
Major
Figure 1 resembles too much the figure in Akimoto J. Photodynamic Therapy for Malignant Brain Tumors. Neurologia medico-chirurgica. 2016;56(4):151-7, which is not cited in the manuscript. I would not say this is a plagiarism but might be suspected. I would recommend modification of the figure or making up from scratch, so that the figure can convey the true meaning that authors may want to show.
Thanks for pointing out. Paper cited and mentioned in the legend to Figure 1. We did borrow some elements which depicts the activation from that review but now the style is different and we think overall the message is much clearer.
Line 250; It is reasonable that the authors focuses on mitochondrial effect of PDT if they specialize in the field. Still, if this is a review for PDT of GBM, the use of talaporfin sodium could be considered to be described, because this is the only government-approved PDT for GBM so far.
Text changed extensively, please see areas highlighted red.
Line 274-283; This summary well describes the reason for need for PDT, but there is too little about PDT itself. Some key text should be added, at least to match the abstract.
Summary now ends with this statement, the description of PDT is provided in many parts of the text above.
Although surgical techniques such as fluorescence-guided surgery has allowed intra-operative real-time visualization of tumour cells, infiltrating GBM cells cannot be mechanically removed, in most cases, because they spread into areas which are too risky to surgically destroy, for example critical white matter tracts. Therefore, the task for the future is to develop effective strategies to cleanse the infiltrating GBM cells from the parenchyma immediately adjacent to the surgical wound. It is here PDT could find its niche.
Minor
Title; as the authors focused very much on the function of mitochondria, this point could be included in the title, so that the potential readers who are particularly interested in those function in (and relationship to) PDT.
We agree but since this idea is yet to be explored in GBM we do not think we have an easy way of doing so.
Line 17; in explanation of photosensitizer, its specific accumulation to tumor should be described, although referring to molecular finger print is also important. (“not specific to particular molecular fingerprint” could be misleading, although correct, if the readers are expecting its particular accumulation to cancer cells.):
Addressed, rephrased
Line 33; the description about grading system in WHO criteria are not correct and needs citation, which is later cited well. The criteria were determined to match clinical outcome. Also, grade 1 glioma should be included in low grade glioma (LGG).
Corrected
Table 1, Row 1 (for Perria et al); “Not defineds” probably a typo.
CORRECTED
Table 1, Last row; citation is missing for Akimoto el al. not missing it is there!
CORRECTED
Line 139; surgery “by” Stummer should be correct.
CORRECTED
Line 140; “Enhancing tumor” might be difficult to understand for researchers. MRI enhancement effect should be mentioned briefly:
REMOVED and REPHRASED
Line 141; the comma between survival and currently should be a period instead.
CORRECTED
Also, FDA approval of 5-ALA was rather late compared to the status in Europe and Japan. This fact should be mentioned.
We do not see how we can fit it into the story or why this would be critical
Line 245; In terms of clinical trial using 5-ALA for PDT, INDYGO is not the only one. Interstitial PDT should be mentioned as a review. Please check “PD L 506 for Stereotactic Interstitial Photodynamic Therapy of Newly Diagnosed Supratentorial IDH Wild-type Glioblastoma”.
We have extensively cited papers on the interstitial PDT and mentioned this trial:
Another trial with 5-ALA, which is currently registered at ClinicalTrials.gov (NCT03897491) by Photonamic GmbH&Co.KG, “PD L 506 for Stereotactic Interstitial Photodynamic Therapy of Newly Diagnosed Supratentorial IDH Wild-type Glioblastoma” is not yet recruiting.
Line 268-271; This part is critically important for clinical application of PDT. Since the authors have made Technical aspect subheading, this description could be moved there so to help readers understand the situation. The following future prospective are reasonable to be described here.
This part has been almost completely rewritten and rearranged, highlighted red.
Reviewer 3 Report
In their manuscript Vasilev et al. review literature on PDT in GBM. The article is of interest for a neurosurgical / neurooncological community and the reference list is adequate for the purpose of the review. I have the following specific comments for the authors to further improve their work.
General:
- The manuscript requires proofreading since there are some typos and inconsistencies included (e.g., key words: “glioblastoma multiform” should rather read as “glioblastoma multiforme”).
Abstract:
- “Surgical attempts to eradicate GBM fail because…”: I suggest that the authors use rather less “radical” wording here as surgical resection of GBM is still the primary treatment of choice in most cases (depending on tumor location and patient demands, of course). Could PDT rather be set in focus as complementary to other treatment options or as an alternative among therapy approaches?
- “… be possible to relatively selectively damage infiltrating GBM cells…”: This is rather vague. Please consider rephrasing.
- “…appropriate photosensitiser can be developed and sufficient light delivered into the tissue.”: Most readers might not yet be familiar with the techniques and principles of PDT. Please consider defining “photosensitiser” and “sufficient light”.
- The abstract may end with an objective or aim of the review.
Introduction:
- Last sentence of the first paragraph: There are some references given for some factors leading to tumor development, but this is inconsistent and not done for all listed factors. Could the authors please expand and provide suitable references for all mentioned factors here?
- Second paragraph: Please consider also mentioning molecular factors associated with glioma grading. This paragraph should be expanded and may introduce other features considered for distinct classification and phenotyping. This paragraph should then be seen as a general introduction to grading (whereas more detailed descriptions should then be given for GBM in the following section where details are currently mixed for LGG and HGG despite the article seems to focus on GBM).
The problem:
- “In all cases, residual growth within critical regions such as white matter tracts, cannot be surgically removed due to risk of severe neurological damage”: I disagree with this statement. I think most tumors have parts that are located within white matter tracts. It seems rather relevant if or for what function a specific white matter tract is essential.
- “…particularly where options for surgical resection are limited.”: Does this apply for initial resection as well as re-resection? In general, what is the role of PDT or does it differ between initial diagnoses and later progression / relapse?
Development of the concept of PDT:
- “The first obvious answer is that PDT, in principle, may effectively cure some cancers, based on results of experiments in-vitro, in rodent models, and clinical studies”: The authors should expand and need to carefully provide references here.
- Could the authors please clarify whether studies reported in Table 1 were in-vitro or in-vivo?
Challenges for development of PDT for GBM:
- While the most urgent challenges become clear, I would suggest that the authors also include – probably as a new, additional paragraph – explicitly formulated limitations in current PDT studies to allow for critical judgement and new ideas to further develop PDT.
Conclusions and perspectives:
- “In summary, GBM remains one of the most dangerous cancers known and there is an acute 274 need for new methods of therapy.” and following sentences: I suggest that the authors carefully revise their conclusions and make them more specific to their review and PDT in GBM.
- The authors may consider splitting conclusions and perspectives in two separate parts, with the conclusions section forming the end paragraph of the review.
Author Response
We would like to thank all the reviewers and the academic editor for their extremely helpful criticisms and suggestions. We have made multiple extensive changes and amendments and hope our revision is satisfactory.
Below we cite the criticisms and provide responses (bold) and extracts from the new text in red.
General:
- The manuscript requires proofreading since there are some typos and inconsistencies included (e.g., key words: “glioblastoma multiform” should rather read as “glioblastoma multiforme”).
CORRECTED
Abstract:
- “Surgical attempts to eradicate GBM fail because…”: I suggest that the authors use rather less “radical” wording here as surgical resection of GBM is still the primary treatment of choice in most cases (depending on tumor location and patient demands, of course). Could PDT rather be set in focus as complementary to other treatment options or as an alternative among therapy approaches?
Revised and changed in line with this request. Text now reads:
The introduction of this protocol has improved overall survival, however recurrence is essentially inevitable. The key reason for that is that the surgical treatment fails to eradicate GBM cells completely, and adjacent parenchyma remains infiltrated by scattered GBM cells which become the source of recurrence. This stimulates interest to any supplementary methods which could help to destroy residual GBM cells and fight the infiltration.
- “… be possible to relatively selectively damage infiltrating GBM cells…”: This is rather vague. Please consider rephrasing.
Changed to: This gives hope that it might be possible to preferentially damage infiltrating GBM cells within the areas which cannot be surgically removed and further improve the chances of survival if an efficient photosensitiser and hardware for light delivery into the brain tissue are developed.
- “…appropriate photosensitiser can be developed and sufficient light delivered into the tissue.”: Most readers might not yet be familiar with the techniques and principles of PDT. Please consider defining “photosensitiser” and “sufficient light”.
We thought how this can be done but in the abstract we do not have much room for extensive explanations. We have rather extensively changed the Abstract, hopefully it reads better now.
- The abstract may end with an objective or aim of the review.
Thank you for this suggestion. We have added:
In this review we discuss the idea that other types of molecules which build up in mitochondria could be explored as photosensitizers and used for PDT of these aggressive brain tumours.
Introduction:
- Last sentence of the first paragraph: There are some references given for some factors leading to tumor development, but this is inconsistent and not done for all listed factors. Could the authors please expand and provide suitable references for all mentioned factors here?
We have removed much of this text because it is not contributing to the rest of the discussion much.
- Second paragraph: Please consider also mentioning molecular factors associated with glioma grading. This paragraph should be expanded and may introduce other features considered for distinct classification and phenotyping. This paragraph should then be seen as a general introduction to grading (whereas more detailed descriptions should then be given for GBM in the following section where details are currently mixed for LGG and HGG despite the article seems to focus on GBM).
We have deleted the last paragraph as it seems to go outside of the remit of the review anyway.
The problem:
- “In all cases, residual growth within critical regions such as white matter tracts, cannot be surgically removed due to risk of severe neurological damage”: I disagree with this statement. I think most tumors have parts that are located within white matter tracts. It seems rather relevant if or for what function a specific white matter tract is essential.
We have re-phrased to:
Essentially in all cases, residual growth within critical regions such as critical white matter tracts or grey matter nuclei, cannot be surgically removed without a risk of severe neurological damage. In order to prevent or impair tumour recurrence and improve patient survival rates, it would be necessary to clear the cavity and the surrounding tissue. This could improve survival for patients but calls for a different approach, such as photodynamic therapy (PDT), using specific sensitising agents termed photosensitisers (PS).
- “…particularly where options for surgical resection are limited.”: Does this apply for initial resection as well as re-resection? In general, what is the role of PDT or does it differ between initial diagnoses and later progression / relapse?
We think that this issue is relevant for both, primary affect and relapse but to simplify, we have removed that sentence, it was a bit redundant anyway.
Development of the concept of PDT:
- “The first obvious answer is that PDT, in principle, may effectively cure some cancers, based on results of experiments in-vitro, in rodent models, and clinical studies”: The authors should expand and need to carefully provide references here.
- Could the authors please clarify whether studies reported in Table 1 were in-vitro or in-vivo?
INCLUDED IN THE LEGEND. ALL IN VIVO
The beginning of that part now reads:
Conceptually, PDT is the use of a chemical which is able to absorb photons in order to trigger a process, detrimental for the tumour cells. Why should development of PDT for GBM be considered? The first obvious answer is that PDT, in principle, may be effective against some cancers, based on results of experiments in-vitro, rodent models, and clinical studies. Examples of cancers, where encouraging results with PDT were obtained in-vitro include breast cancer [11], oesophageal cancer[12] , liver carcinoma [13], and colorectal cancer [14]. Further information on in-vivo application of PDT can be found elsewhere [15].
We have indicated in the text whether studies are in vitro or in vivo and in the Table legend, that all mentioned studies are in vivo.
Challenges for development of PDT for GBM:
- While the most urgent challenges become clear, I would suggest that the authors also include – probably as a new, additional paragraph – explicitly formulated limitations in current PDT studies to allow for critical judgement and new ideas to further develop PDT.
We have now included the following text:
From the available literature, is appears that so far, the efficacy of porphyrins as exciters for PDT remains uncertain, although they greatly aid the detection of tumour margins. In fact, it is hard to distinguish whether the damage to the tumour caused in the animal or human experiments was due to the porphyrins at all. For example, in Olzowy et al. experiments [38], red (633 nm) light caused the same degree of damage irrespective of the loading with 5-ALA. It seems that a different combination of the photosensitiser and wavelength could produce better results, possibly via not focusing only on selective biotransformation of 5ALA in the GBM but exploiting some other PS and features of the tumour cells.
The other area of important development is the hardware for light delivery. The important key steps have already been made by introduction of the interstitial PDT. This method seems to be the only feasible strategy to deliver light to the GBM cells underneath the surface of the cavity left after the debulking. Stereotaxic placement of the arrays of special light diffusers may help to illuminate the required volume of the parenchyma without causing dangerous mechanical damage to the surrounding tissue.
Conclusions and perspectives:
- “In summary, GBM remains one of the most dangerous cancers known and there is an acute 274 need for new methods of therapy.” and following sentences: I suggest that the authors carefully revise their conclusions and make them more specific to their review and PDT in GBM.
Conclusions and perspectives are now separate, we revised the text to remove mistakes
- The authors may consider splitting conclusions and perspectives in two separate parts, with the conclusions section forming the end paragraph of the review.
We have split them as requested.
Round 2
Reviewer 3 Report
The authors have largely implemented my comments during the first round of revision. I have no further comments.